# Synthesizing Robust Adversarial Examples

## Abstract

Neural network-based classifiers parallel or exceed human-level accuracy on many common tasks and are used in practical systems. Yet, neural networks are susceptible to adversarial examples, carefully perturbed inputs that cause networks to misbehave in arbitrarily chosen ways. When generated with standard methods, these examples do not consistently fool a classifier in the physical world due to a combination of viewpoint shifts, camera noise, and other natural transformations. Adversarial examples generated using standard techniques require complete control over direct input to the classifier, which is impossible in many real-world systems.

We introduce the first method for constructing real-world 3D objects that consistently fool a neural network across a wide distribution of angles and viewpoints. We present a general-purpose algorithm for generating adversarial examples that are robust across any chosen distribution of transformations. We demonstrate its application in two dimensions, producing adversarial images that are robust to noise, distortion, and affine transformation. Finally, we apply the algorithm to produce arbitrary physical 3D-printed adversarial objects, demonstrating that our approach works end-to-end in the real world. Our results show that adversarial examples are a practical concern for real-world systems.

## 1 Introduction

The existence of adversarial examples for neural networks has until now been largely a theoretical concern. While minute, carefully-crafted perturbations can cause targeted misclassification in a neural network, adversarial examples produced using standard techniques lose adversariality when directly translated to the physical world as they are captured over varying viewpoints and affected by natural phenomena such as lighting and camera noise. This suggests that practical systems may not be at risk because adversarial examples generated using standard techniques are not robust in the physical world.

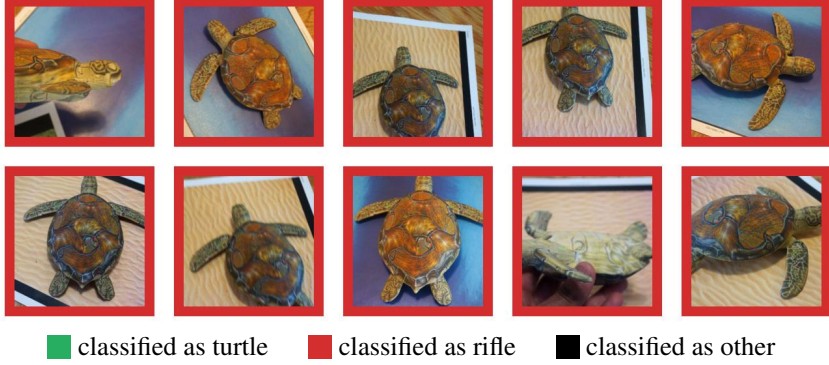

■ classified as turtle    ■ classified as rifle    ■ classified as other

Figure 1: Randomly sampled poses of a **single** 3D-printed turtle adversarially perturbed to classify as a rifle at every viewpoint by an ImageNet classifier. An unperturbed model is classified correctly as a turtle 100% of the time. See `https://youtu.be/YXy6oX1iNoA` for a video where every frame is fed through the classifier: the turtle is consistently classified as a rifle.

We show that neural network-based classifiers are vulnerable to physical-world adversarial examples. We introduce a new algorithm for reliably producing physical 3D objects that are adversarial over a distribution of viewpoints. Figure 1 shows an example of an adversarial object constructed using our approach, where a 3D-printed turtle is consistently classified as rifle by an ImageNet classifier. In this paper, we demonstrate the efficacy and generality of our method, demonstrating conclusively that adversarial examples are a concern in real-world systems.

## 1.1 CHALLENGES

Methods for transforming ordinary two-dimensional images into adversarial examples are well-known; however, the extent to which these techniques affect specific real-world systems is still an open question. Prior work has shown that adversarial examples generated using standard techniques lose their adversarial nature once subjected to minor transformations (Luo et al., 2016; Lu et al., 2017). These results suggest that adversarial examples, while serving as interesting phenomena, have no hope of applying to physical-world systems, where transformations such as viewpoint changes and camera noise are inevitable.

Prior techniques attempting to synthesize robust adversarial examples for the physical world have had limited success. While some progress has been made, current efforts have demonstrated a small number of datapoints on nonstandard classifiers, and only in the two-dimensional case, with no clear generalization to three dimensions (further discussed in Section 4).

The entirety of prior work has only explored generating adversarial examples in the two-dimensional case, where "viewpoints" can be approximated by an affine transformations of an original image. Constructing adversarial examples for the physical world requires the ability to generate entire 3D adversarial objects, which must remain adversarial in the face of complex transformations not applicable to 2D objects, such as 3D rotations and perspective projection.

## 1.2 CONTRIBUTIONS

In this work, we definitively show that adversarial examples pose a real threat in the physical world. We propose a general-purpose algorithm for reliably constructing adversarial examples robust over a chosen distribution of transformations, and we demonstrate the efficacy of this algorithm in both the 2D and 3D case. We succeed in producing physical-world 3D adversarial objects that are robust over a large, realistic distribution of 3D viewpoints, proving that the algorithm produces adversarial three-dimensional objects that are adversarial in the physical world. Specifically, our contributions are as follows:

- We develop Expectation Over Transformation (EOT), a novel algorithm that produces single adversarial examples that are simultaneously adversarial over an entire distribution of transformations

- We consider the problem of constructing 3D adversarial examples under the EOT framework, viewing the 3D rendering process as part of the transformation, and we show that the approach successfully synthesizes adversarial objects

- We fabricate adversarial objects and show that they remain adversarial, demonstrating that our approach works end-to-end in the physical world, showing that adversarial examples are of real concern in practical deep learning systems

## 2 APPROACH

First, we present the Expectation Over Transformation (EOT) algorithm, a general framework allowing for the construction of adversarial examples that remain adversarial over a chosen transformation distribution $T$. We then describe our end-to-end approach for generating adversarial objects using a specialized application of EOT in conjunction with differentiating through the 3D rendering process.

## 2.1 EXPECTATION OVER TRANSFORMATION

When constructing adversarial examples in the white-box case (that is, with access to a classifier and its gradient), we know in advance a set of possible classes $Y$ and a space of valid inputs $X$ to the classifier; we have access to the function $P(y|x)$ and its gradient $\nabla P(y|x)$, for any class $y \in Y$ and input $x \in X$. In the standard case, adversarial examples are produced by maximizing the log-likelihood of the target class over a $\epsilon$-radius ball around the original image, that is:

$$\hat{x} = \arg\max_{x' \in X} \log P(y|x') \qquad \text{s.t. } ||x' - x||_p < \epsilon$$

This approach has been shown to be both feasible and effective at generating adversarial examples for any given classifier. However, prior work has shown adversarial examples' inability to remain adversarial even under minor perturbations inevitable in any real-world observation (Luo et al., 2016; Lu et al., 2017).

To address this issue, we introduce *Expectation Over Transformation (EOT)*. The key insight behind EOT is to model such perturbations within the optimization procedure. Rather than optimizing the log-likelihood of a single example, EOT uses a chosen distribution $T$ of transformation functions $t$ taking an input $x'$ supplied by the adversary to the "true" input $t(x')$ perceived by the classifier. Furthermore, rather than simply taking the norm of $x' - x$ to constrain the solution space, EOT instead aims to constrain the *effective distance* between the adversarial and original inputs, which we define as:

$$\delta = \mathbb{E}_{t \sim T}[d(t(x) - t(x'))]$$

Intuitively, this is how different we expect the true input to the classifer will be, given our new input. Then, EOT solves the following optimization problem:

$$\hat{x} = \arg\max_{x'} \mathbb{E}_{t \sim T}[\log P(y|t(x'))] \qquad \text{s.t. } \mathbb{E}_{t \sim T}[d(t(x'), t(x))] < \epsilon$$

In practice, the distribution $T$ can model perceptual distortions such as random rotation, translation, or addition of noise. However, EOT generalizes beyond simple transformations. EOT finds examples robust under any perception distribution $Q(\cdot \, ; x)$ parameterized by the generated example $x$ as long as $\frac{d}{dx}Q(\cdot \, ; x)$ is well-defined. The objective function can be optimized by stochastic gradient descent, approximating the derivative of the expected value through sampling transformations independently at each gradient descent step and differentiating through the transformation.

## 2.2 SYSTEM OVERVIEW

Given its ability to synthesize robust adversarial examples, we use the EOT framework for generating 2D examples, 3D models, and ultimately physical-world adversarial objects. Within the framework, however, there is a great deal of freedom in the actual method by which examples are generated, including choice of $T$, distance metric, and optimization method.

In the 2D case, we design $T$ to approximate a realistic space of possible distortions involved in printing out an image and taking a natural picture of it. This amounts to a set of random transformations of the form $t(x) = Ax + \epsilon$, which are more thoroughly described in Section 3.

In the 3D case, we define the transformations $t$ from a *texture* to a perceived image by mapping the texture to a given 3D object, and simulating highly nonlinear functions such as rendering, rotation, translation, and perspective distortion of the object in addition to the perceptual mechanisms used in the 2D case.

Finding adversarial examples across these nonlinear transformations is what allows for the transfer of adversarial examples to the physical world, but also introduces implementation complexities not found in the 2D case. EOT requires the ability to differentiate though the transformation distribution with respect to the input. In the 3D case, this implies differentiating through the 3D renderer with respect to the texture.

To do this, we model the rendering process as a sparse matrix multiplication between the texture and a coordinate map, which is generated from a standard 3D renderer. Rather than attempting to differentiate through the complex renderer, we use the renderer to find a coordinate map for each

rendering, which maps coordinates on the texture to coordinates in the classifier's field of view. We can then simulate this specific rendering as a sparse matrix multiplication and differentiate through it; we differentiate through different matrix multiplications at each sampling step.

Once EOT has been parameterized, i.e. once a distribution $T$ is chosen, the issue of actually optimizing the induced objective function remains. Rather than solving the constrained optimization problem given above, we use the Lagrangian-relaxed form of the problem, as Carlini & Wagner (2017) do in the conventional (non-EOT, single-viewpoint) case:

$$\hat{x} = \arg\min_{x'} \mathbb{E}_{t \sim T}[-\log P(y|t(x'))] + \lambda \mathbb{E}_{t \sim T}[d(t(x'), t(x))]$$

In order to encourage imperceptibility of the generated images, we set $d(x', x)$ to be the $\ell_2$ norm in the LAB color space, a perceptually uniform color space where Euclidean distance roughly corresponds with perceptual distance. Note that the $\mathbb{E}_{t \sim T}[||LAB(t(x)) - LAB(t(\hat{x}))||_2^2]$ can be sampled and estimated in conjunction with $\mathbb{E}[P(y|t(x))]$; in general, the Lagrangian formulation gives EOT the ability to intricately constrain the search space (in our case, using LAB distance) at insignificant computational cost (without computing a complex projection). Our optimization, then, is:

$$\hat{x} = \arg\min_{x'} \mathbb{E}_{t \sim T}[-\log P(y|t(x')) + \lambda ||LAB(t(x)) - LAB(t(x'))||_2^2]$$

We use projected gradient descent to find the optimum, clipping to the set of valid inputs (e.g. $[0, 1]$ for images).

## 3 EVALUATION

We show that we can reliably produce transformation-tolerant adversarial examples in both the 2D and 3D case. Furthermore, we show that we can synthesize and fabricate 3D adversarial objects, even those with complex shapes, in the physical world: these adversarial objects remain adversarial regardless of viewpoint, camera noise, and other similar real-world factors. *Adversariality* in this case refers to the propensity of the classifier to predict an attacker-chosen target class given an image or object.

In our evaluation, given a source object $x$ and a set of correct classes $\{y_1, \ldots y_n\}$, as well as an attacker-chosen target class $y_{adv}$ and a crafted adversarial example $x'$, we use the following terms in order to characterize the effectiveness of $x'$. We call a viewpoint "adversarial" if the top output of the classifier is $y_{adv}$; "correct" if the adversarial example fails and the classifier outputs one of $\{y_1, \ldots, y_n\}$, and "misclassified" if the classifier predicts a class unrelated to that of the source object (i.e. not in $\{y_1, \ldots y_n\}$) but not the target class $y_{adv}$. By randomly sampling a set of viewpoints, we can evaluate an adversarial example $x'$ by examining the proportion of adversarial, correct, and misclassified viewpoints.

We evaluate robust adversarial examples in the 2D and 3D case, and furthermore, we evaluate physical-world 3D adversarial objects. The two cases are fundamentally different. In the virtual case, we know that we want to construct adversarial examples robust over a certain distribution of transformations, and we can simply use EOT over that distribution to synthesize a robust adversarial example. In the case of the physical world, however, we cannot capture the exact distribution unless we perfectly model all physical phenomena. Therefore, we must approximate the distribution and perform EOT over the proxy distribution. This works well in practice for producing adversarial objects that remain adversarial under the "true" physical-world distribution, as we demonstrate. See Appendix A for the exact parameters of the distributions we use in the 2D, 3D simulation, and 3D physical-world cases. In all cases, the various transformation parameters are sampled as continuous random variables from a uniform distribution between the minimum and maximum values given, unless otherwise indicated in Appendix A (i.e. Gaussian noise).

In our experiments, we use TensorFlow's standard pre-trained InceptionV3 classifier (Szegedy et al., 2015) which has 78.0% top-1 accuracy on ImageNet. In all of our experiments, we use randomly chosen target classes, and we use EOT to synthesize adversarial examples over a chosen distribution. We measure the $\ell_2$ distance per pixel between the original and adversarial example (in LAB

| Images | Classification Accuracy | | Adversariality | | $\ell_2$ |
|---|---|---|---|---|---|
| | mean | stdev | mean | stdev | mean |
| Original | 70.0% | 36.4% | 0.01% | 0.3% | N/A |
| Adversarial | 0.9% | 2.0% | 96.4% | 4.4% | $5.6 \times 10^{-5}$ |

Table 1: Evaluation of 1000 2D adversarial examples with random targets. We evaluate each example over 1000 randomly sampled transformations to calculate classification accuracy and adversariality (percent classified as the adversarial class).

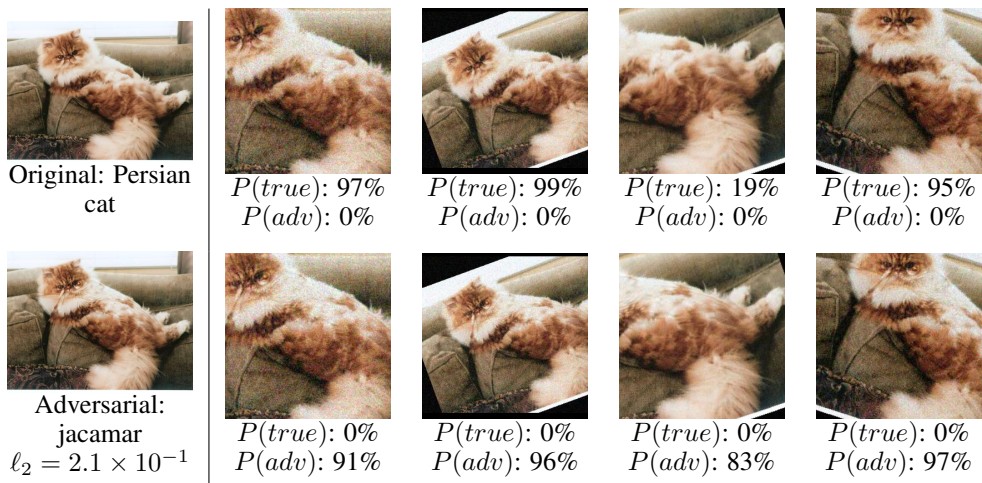

Original: Persian cat

$P(true)$: 97% $P(adv)$: 0%   $P(true)$: 99% $P(adv)$: 0%   $P(true)$: 19% $P(adv)$: 0%   $P(true)$: 95% $P(adv)$: 0%

Adversarial: jacamar
$\ell_2 = 2.1 \times 10^{-1}$

$P(true)$: 0% $P(adv)$: 91%   $P(true)$: 0% $P(adv)$: 96%   $P(true)$: 0% $P(adv)$: 83%   $P(true)$: 0% $P(adv)$: 97%

Figure 2: A 2D adversarial example, showing classifier confidence in true and adversarial classes for original and corresponding adversarial image over randomly sampled poses.

space), and we also measure classification accuracy (percent of randomly sampled viewpoints classified as the true class) and adversariality (percent of randomly sampled viewpoints classified as the adversarial class) for both the original and adversarial example. When working in simulation, we evaluate over a large number of transformations sampled randomly from the distribution; in the physical world, we evaluate over a large number of manually-captured images of our adversarial objects taken over different viewpoints.

### 3.1 ROBUST 2D ADVERSARIAL EXAMPLES

In the 2D case, we consider the distribution of transformations that includes rescaling, rotation, lightening or darkening by an additive factor, adding Gaussian noise, and any in-bounds translation of the image.

We take the first 1000 images in the ImageNet validation set, randomly choose a target class for each image, and use EOT to synthesize an adversarial example that is robust over the chosen distribution. For each adversarial example, we evaluate over 1000 random transformations sampled from the distribution at evaluation time. Table 1 summarizes the results. On average, the adversarial examples have an adversariality of 96.4%, showing that our approach is highly effective in producing robust adversarial examples. Figure 2 shows an example of a synthesized adversarial example, along with the classification confidence in true and adversarial classes for original and corresponding adversarial images. See Appendix B for more examples.

### 3.2 ROBUST 3D ADVERSARIAL EXAMPLES

We produce 3D adversarial examples by modeling the 3D rendering as a transformation under EOT. Given a textured 3D object, we optimize over the texture such that the rendering is adversarial from any viewpoint. We consider a distribution that incorporates different camera distances, lateral trans-

| Images | Classification Accuracy | | Adversariality | | $\ell_2$ |
|---|---|---|---|---|---|
| | mean | stdev | mean | stdev | mean |
| Original | 68.8% | 31.2% | 0.01% | 0.1% | N/A |
| Adversarial | 1.1% | 3.1% | 83.4% | 21.7% | $5.9 \times 10^{-3}$ |

Table 2: Evaluation of 200 3D adversarial examples with random targets. We evaluate each example over 100 randomly sampled poses to calculate classification accuracy and adversariality (percent classified as the adversarial class).

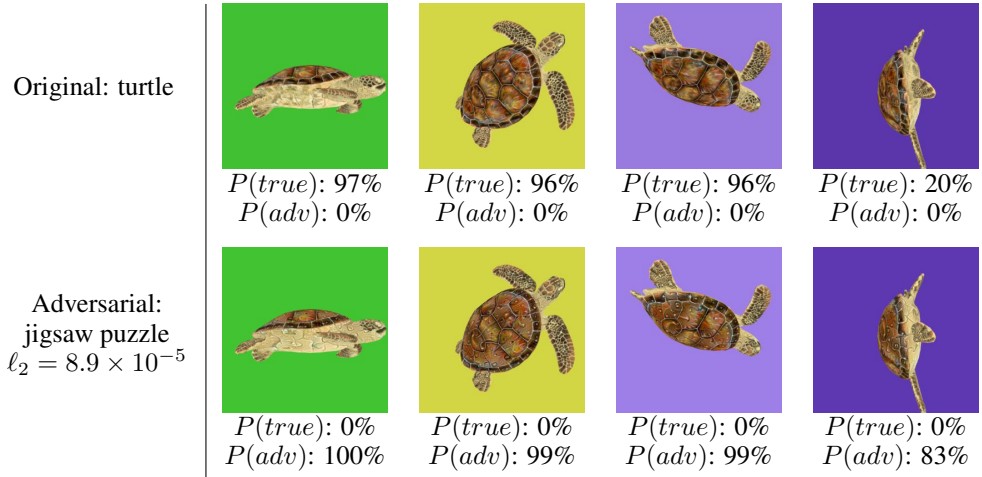

Figure 3: A 3D adversarial example, showing classifier confidence in true and adversarial classes for original and corresponding adversarial object over randomly sampled poses.

lation, rotation of the object, and solid background colors. We approximate the expectation over transformation by taking the mean loss over batches of size 40; furthermore, due to the computational expense of computing a batch, we reuse up to 80% of the batch at each iteration, but enforce that each batch contain at least 8 new images. As previously mentioned, the parameters of the distribution we use is specified in Appendix A, sampled as independent continuous random variables (that are uniform except for Gaussian noise).

We consider 10 complex 3D models, choose 20 random target classes per 3D model, and use EOT to synthesize adversarial textures for the 3D models with minimal parameter search (four constant, pre-chosen $\lambda$ values were tested across each [3D model, target] pair). For each of the 200 adversarial examples, we sample 100 random transformations from the distribution at evaluation time. Table 2 summarizes results, and Figure 3 shows renderings of drawn samples with classification probabilities. See Appendix C for more examples.

The simulated adversarial object have an average adversariality of 83.4% with a long left tail, showing that EOT usually produces highly adversarial objects. See Appendix C for a plot of the distribution.

## 3.3 PHYSICAL ADVERSARIAL EXAMPLES

In order to fabricate physical-world adversarial examples, beyond modeling the 3D rendering process, we need to model physical-world phenomena such as lighting effects and camera noise. Furthermore, we need to model the 3D printing process: in our case, we use commercially available full-color 3D printing. With the 3D printing technology we use, we find that color accuracy varies between prints, so we model printing errors as well. We approximate all of these phenomena by a distribution of transformations under EOT. In addition to the transformations considered for 3D in simulation, we consider camera noise, additive and multiplicative lighting, and per-channel color inaccuracies.

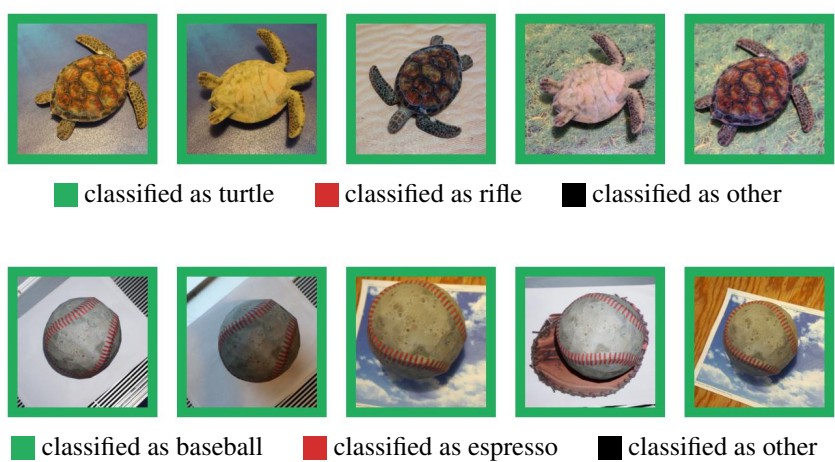

Figure 4: A sample of photos of **unperturbed** 3D prints. The unperturbed 3D-printed objects are consistently classified as the true class.

| Object | Adversarial | Misclassified | Correct |
|---|---|---|---|
| Turtle | 82% | 16% | 2% |
| Baseball | 59% | 31% | 10% |

Table 3: Quantitative analysis of the two adversarial objects, over 100 photos of each object over a wide distribution of viewpoints. Both objects are classified as the adversarial target class in the majority of viewpoints.

We evaluate physical adversarial examples over two 3D-printed objets: one of a turtle (where we consider any of the 5 turtle classes in ImageNet as the "true" class), and one of a baseball. The unperturbed 3D-printed objects are correctly classified as the true class with 100% accuracy over a large number of samples. Figure 4 shows example photographs of unperturbed objects, along with their classifications.

We choose target classes for each of the 3D models at random — "rifle" for the turtle, and "espresso" for the baseball — and we use EOT to synthesize adversarial examples. We evaluate the performance of our two 3D-printed adversarial objects by taking 100 photos of each object over a variety of viewpoints[1]. Figure 5 shows a random sample of these images, along with their classifications. Table 3 gives a quantitative analysis over all images, showing that our 3D-printed adversarial objects are strongly adversarial over a wide distribution of transformations. See Appendix D for more examples.

## 3.4 DISCUSSION

The results and quantative analysis in this section demonstrate the efficacy of EOT and confirm the existence of physical adversarial examples. Here, we perform a qualitative analysis of the results:

**Modeling Perception.** The EOT algorithm as presented in Section 2 presents a general method to construct adversarial examples over a chosen perceptual distribution, but notably gives no guarantees for observations of the image outside of the chosen distribution. In constructing physical-world adversarial objects, we use a crude, high-variance approximation of the rendering and capture process, and this succeeds in ensuring robustness to a diverse set of environments; see, for example, Figure 6, which shows the same adversarial turtle in vastly different environments. In specialized

---

[1]Although the viewpoints were not selected in any way and were simply the result of walking around the objects, moving them up/down, etc., we hesitate to call them "random" since they were not in fact generated numerically or sampled from a concrete distribution, in contrast with the rendered 3D examples.

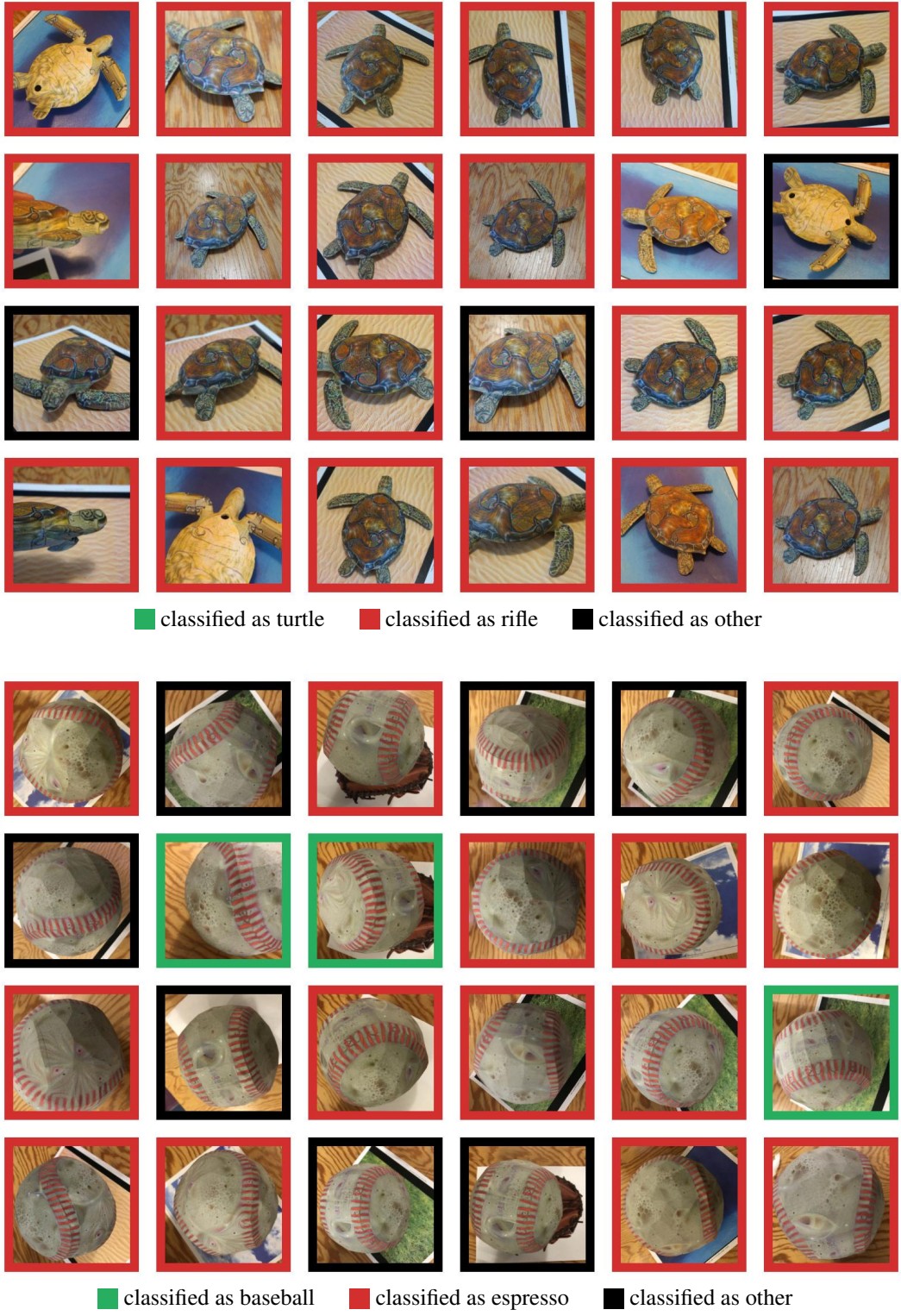

Figure 5: Random sample of photographs of the **two 3D-printed adversarial objects**. The 3D-printed adversarial objects are strongly adversarial over a wide distribution of viewpoints.

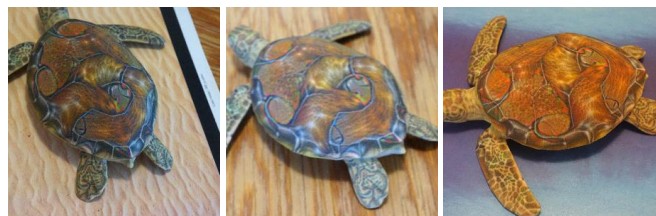

Figure 6: Three pictures of the same adversarial turtle (all classified as "rifle"), demonstrating the need for a wide distribution, and the efficacy of EOT in finding examples robust across wide distributions of physical-world effects like lighting.

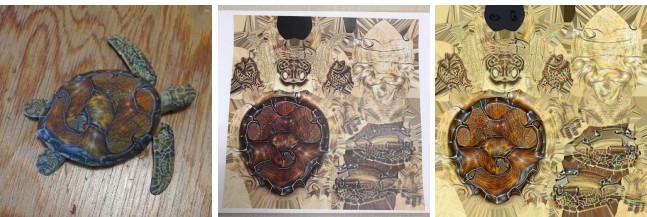

Figure 7: A side-by-side comparison of a 3D-printed model (left) along with a printout of the corresponding texture, printed on a standard laser color printer (center) and the original digital texture (right), showing significant error in color accuracy in printing.

domains, however, a domain expert may opt to model the perceptual distribution precisely in order to better constrain the search space.

**Error in Printing.**  We also find significant error in the color accuracy of even state of the art commercially available color 3D printing; Figure 7 shows a comparison of a 3D-printed model along with a printout of the model's texture, printed on a standard laser color printer. Still, EOT was able to overcome the problem and produce robust physical-world adversarial objects. We predict that we could have produced adversarial examples with smaller $\ell_2$ perturbation with a higher-fidelity printing process.

**Semantically Relevant Misclassification.**  Interestingly, for the majority of viewpoints where the adversarial target class is not the top-1 predicted class, the classifier also fails to correctly predict the source class. Instead, we find that the classifier often classifies the object as an object that is *semantically similar* to the adversarial target; while generating the adversarial turtle to be classified as a rifle, for example, the second most popular class (after "rifle") was "revolver," followed by "holster" and then "assault rifle." Similarly, when generating the baseball to be classified as an espresso, the example was often classified as "coffee" or "bakery".

## 4 RELATED WORK

State of the art neural networks are vulnerable to adversarial examples (Szegedy et al., 2013). Researchers have proposed a number of methods for synthesizing adversarial examples in the white-box scenario (with access to the gradient of the classifier), including L-BFGS (Szegedy et al., 2013), the Fast Gradient Sign Method (FGSM) (Goodfellow et al., 2015), and a Lagrangian relaxation formulation (Carlini & Wagner, 2017), all for the single-viewpoint case where the adversary directly controls the input to the neural network. Projected Gradient Descent (PGD) can be seen as a universal first-order adversary (Madry et al., 2017).

Moosavi-Dezfooli et al. (2017) show the existence of universal (image-agnostic) adversarial perturbations, small perturbation vectors that can be applied to any image to induce misclassification. Their paper proposes an algorithm that finds perturbations that are universal over images; in our work, we give an algorithm that finds a perturbation to a single image or object that is universal over a chosen distribution of transformations. In preliminary experiments, we found that universal

adversarial perturbations, like standard adversarial perturbations to single images, are not inherently robust to transformation.

In the first work on 2D physical-world adversarial examples, Kurakin et al. (2016) demonstrate the transferability of FGSM-generated adversarial misclassification on a printed page. In their setup, a photo is taken of a printed image with QR code guides, and the resultant image is warped, cropped, and resized to become a square of the same size as the source image before classifying it. Their results show the existence of 2D physical-world adversarial examples for approximately axis-aligned views, demonstrating that adversarial perturbations produced using FGSM can translate to the physical world and are robust to camera noise, rescaling, and lighting effects. Kurakin et al. (2016) do not synthesize targeted physical-world adversarial examples, they do not evaluate other real-world 2D transformations such as rotation, skew, translation, or zoom, and their approach does not translate to the 3D case.

Sharif et al. (2016) develop a real-world adversarial attack on a state-of-the-art face recognition algorithm, where adversarial eyeglass frames cause targeted misclassification in portrait photos. The algorithm produces robust perturbations through optimizing over a fixed set of inputs: the attacker collects a set of images and finds a perturbation that minimizes cross entropy loss over the set. The algorithm solves a different problem than we do in our work: it produces adversarial perturbations universal over portrait photos taken head-on from a single viewpoint, while EOT produces 2D/3D adversarial examples robust over transformations. The approach also includes a mechanism for enhancing perturbations' printability using a color map to address the limited color gamut and color inaccuracy of the printer. Note that this differs from our approach in achieving printability: rather than creating a color map, we find an adversarial example that is robust to color inaccuracy. Our approach has the advantage of working in settings where color accuracy varies between prints, as was the case with our 3D-printer.

Recently, Evtimov et al. (2017) proposed a method for generating robust physical-world adversarial examples in the 2D case by optimizing over a fixed set of manually-captured images. However, the approach is limited to the 2D case, with no clear translation to 3D, where there is no simple mapping between what the adversary controls (the texture) and the observed input to the classifier (an image). Furthermore, the approach requires the taking and preprocessing of a large number of photos in order to produce each adversarial example, which may be expensive or even infeasible for many objects.

Lu et al. (2017) argued that adversarial examples may not be a practical concern in physical-world systems because adversarial examples generated for the single viewpoint case lose adversariality at viewpoints with differing scale and rotation. While this is the case with standard adversarial examples as evaluated in the paper and in Luo et al. (2016), we have shown that with EOT, it is in fact possible to construct physical-world adversarial images and objects that are classified as a chosen target class over a wide range of viewpoints.

## 5 CONCLUSION

Our work shows that adversarial examples pose a practical threat to systems using neural network-based image classifiers. By introducing EOT, a general-purpose algorithm for creating robust adversarial examples under any chosen distribution, and modeling 3D rendering and printing within the framework of EOT, we succeed in fabricating three-dimensional adversarial objects. With access only to low-cost commercially available 3D printing technology, we successfully print physical adversarial objects that are strongly classified as a chosen target class over a variety of angles, viewpoints, and lighting conditions by a standard ImageNet classifier.

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

| Transformation | Minimum | Maximum |
|---|---|---|
| Scale | 0.9 | 1.4 |
| Rotation | $-22.5°$ | $22.5°$ |
| Lighten / Darken | $-0.05$ | 0.05 |
| Gaussian Noise (stdev) | 0.0 | 0.1 |
| Translation | any in-bounds | |

Table 4: Distribution of transformations for the 2D case, where each parameter is sampled uniformly at random from the specified range.

| Transformation | Minimum | Maximum |
|---|---|---|
| Camera distance | 2.5 | 3.0 |
| X/Y translation | $-0.05$ | 0.05 |
| Rotation | any | |
| Background | (0.1, 0.1, 0.1) | (1.0, 1.0, 1.0) |

Table 5: Distribution of transformations for the 3D case when working in simulation, where each parameter is sampled uniformly at random from the specified range.

# A    DISTRIBUTIONS OF TRANSFORMATIONS

Under the EOT framework, we must choose a distribution of transformations, and the optimization produces an adversarial example that is robust under the distribution of transformations. Here, we give the specific parameters we chose in the 2D (Table 4), 3D (Table 5), and physical-world case (Table 6).

# B    ROBUST 2D ADVERSARIAL EXAMPLES

We give a random sample out of our 1000 2D adversarial examples in Figures 8 and 9.

# C    ROBUST 3D ADVERSARIAL EXAMPLES

We give a random sample out of our 200 3D adversarial examples in Figures 10 and 11 and 12. We give a histogram of adversariality (percent classified as the adversarial class) over all 200 examples in Figure 13.

| Transformation | Minimum | Maximum |
|---|---|---|
| Camera distance | 2.5 | 3.0 |
| X/Y translation | $-0.05$ | 0.05 |
| Rotation | any | |
| Background | (0.1, 0.1, 0.1) | (1.0, 1.0, 1.0) |
| Lighten / Darken (additive) | $-0.15$ | 0.15 |
| Lighten / Darken (multiplicative) | 0.5 | 2.0 |
| Per-channel (additive) | $-0.15$ | 0.15 |
| Per-channel (multiplicative) | 0.7 | 1.3 |
| Gaussian Noise (stdev) | 0.0 | 0.1 |

Table 6: Distribution of transformations for the physical-world 3D case, approximating rendering, physical-world phenomena, and printing error.

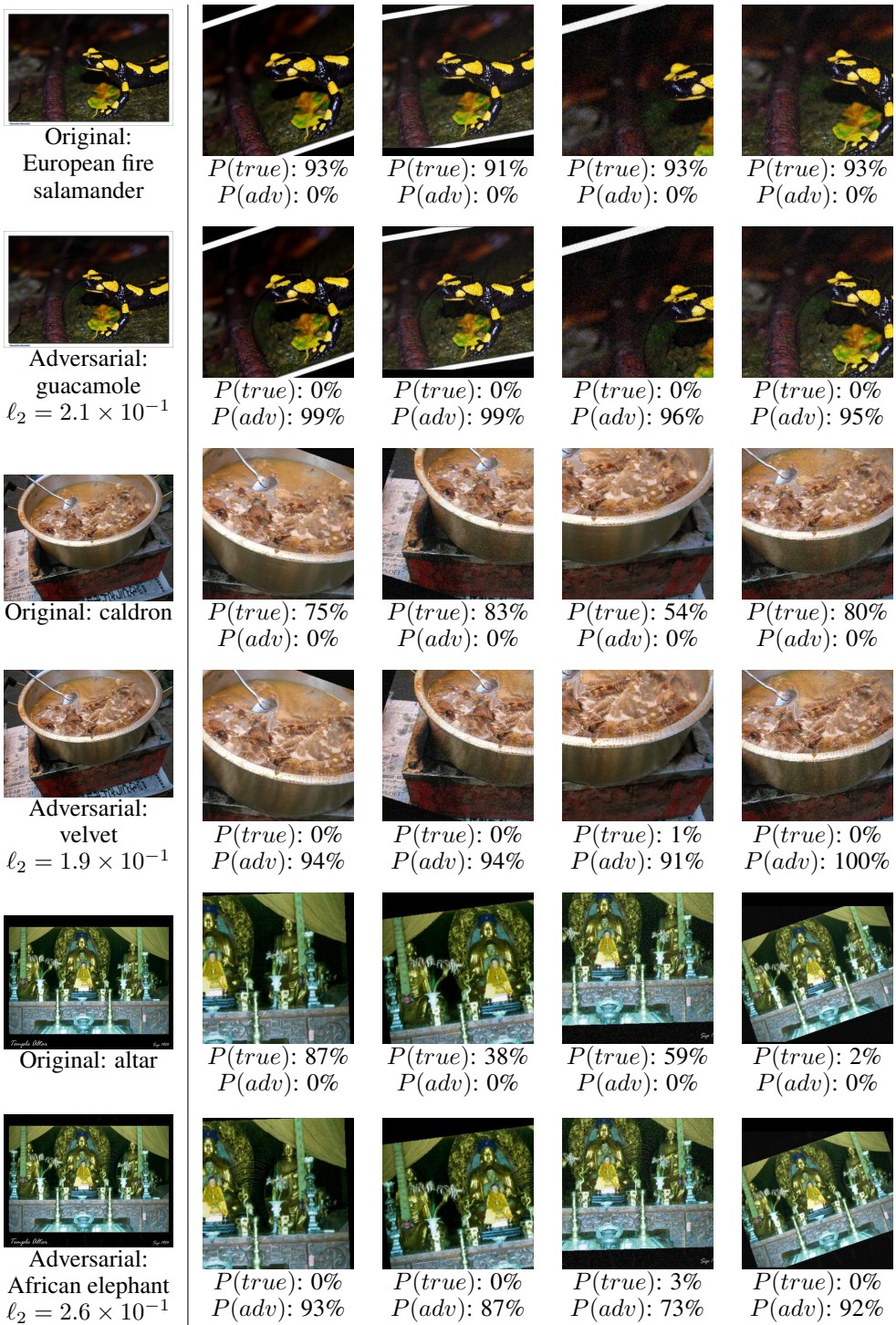

Figure 8: A random sample of 2D adversarial examples.

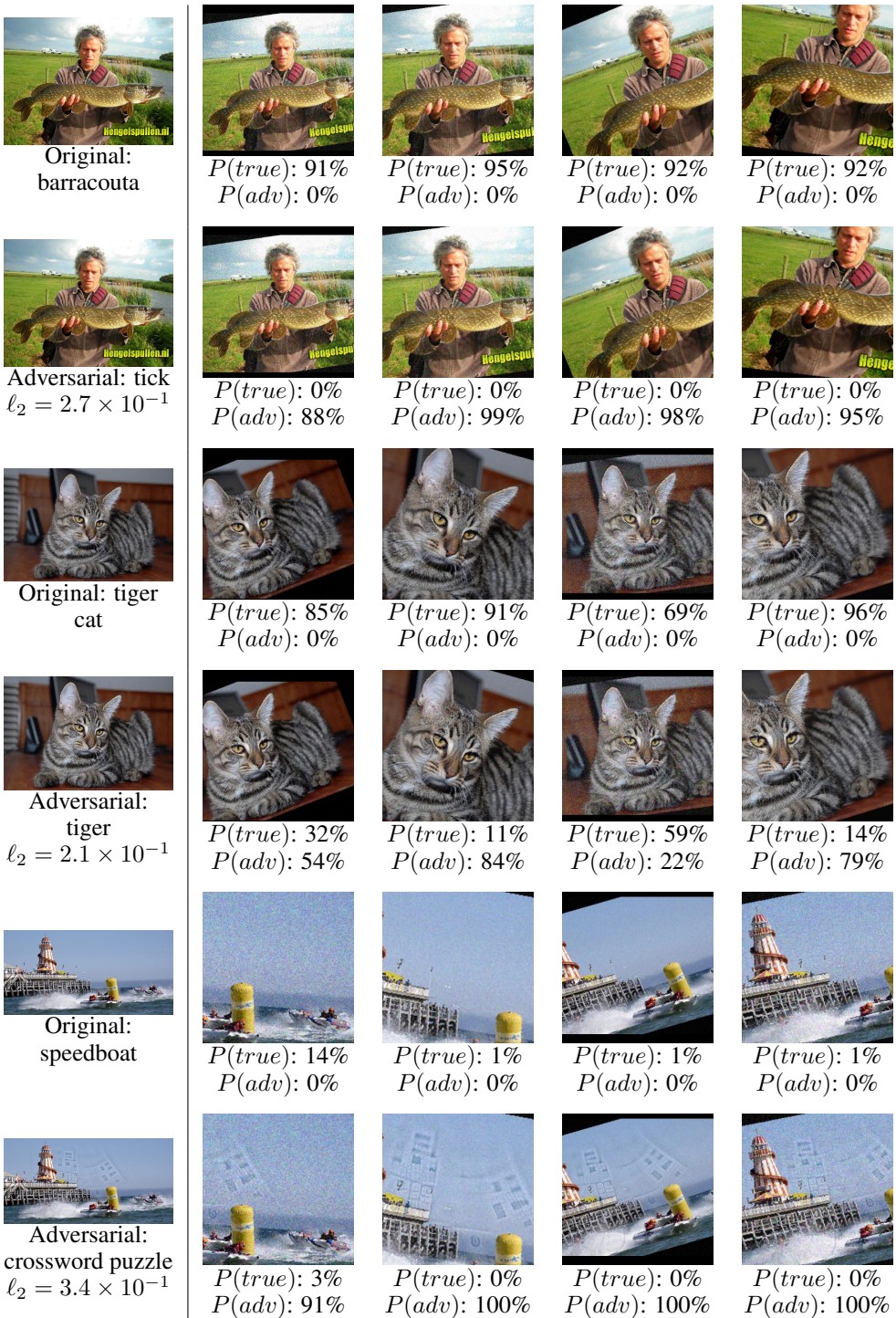

Figure 9: A random sample of 2D adversarial examples.

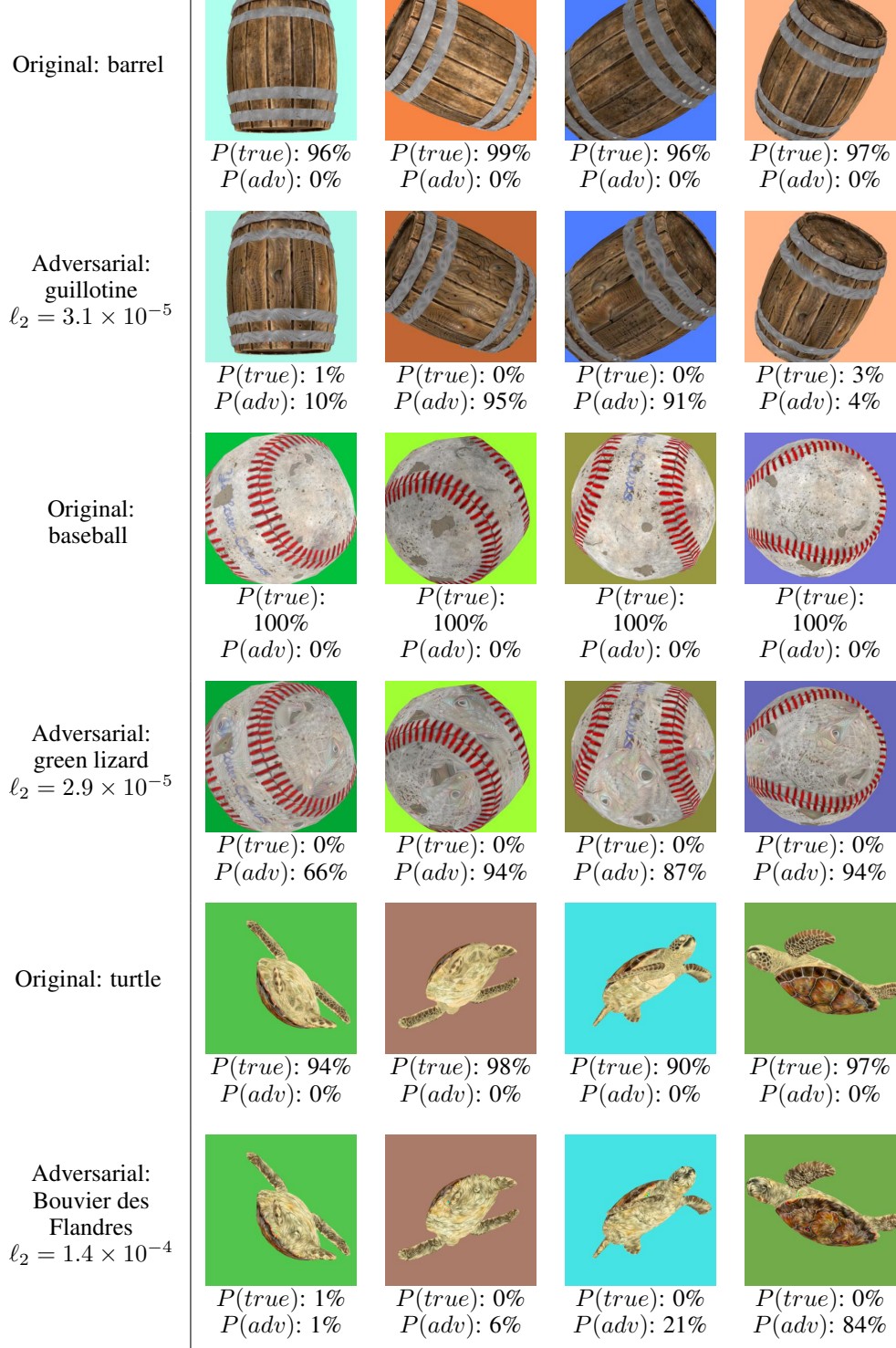

Figure 10: A random sample of 3D adversarial examples.

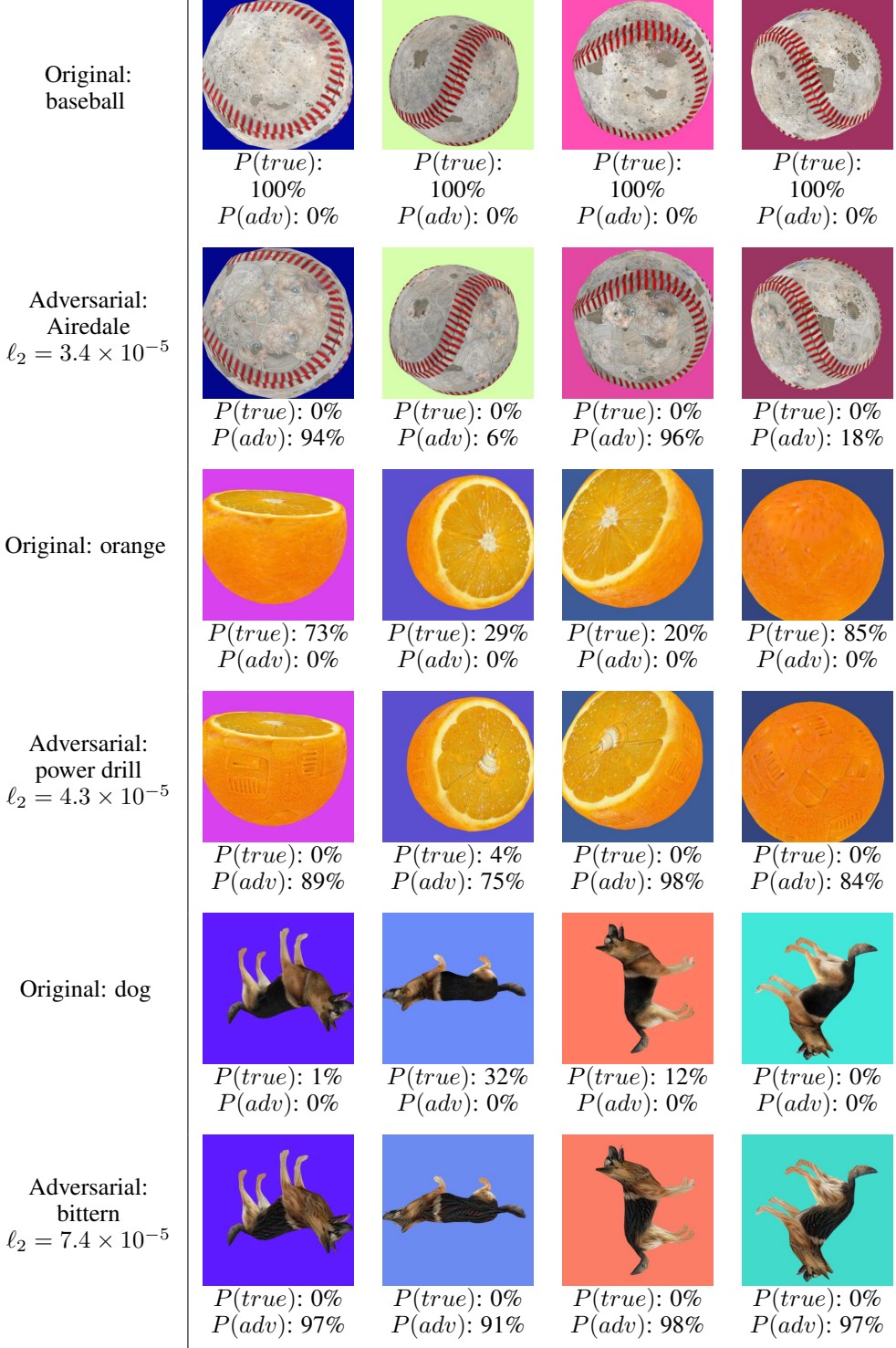

Figure 11: A random sample of 3D adversarial examples.

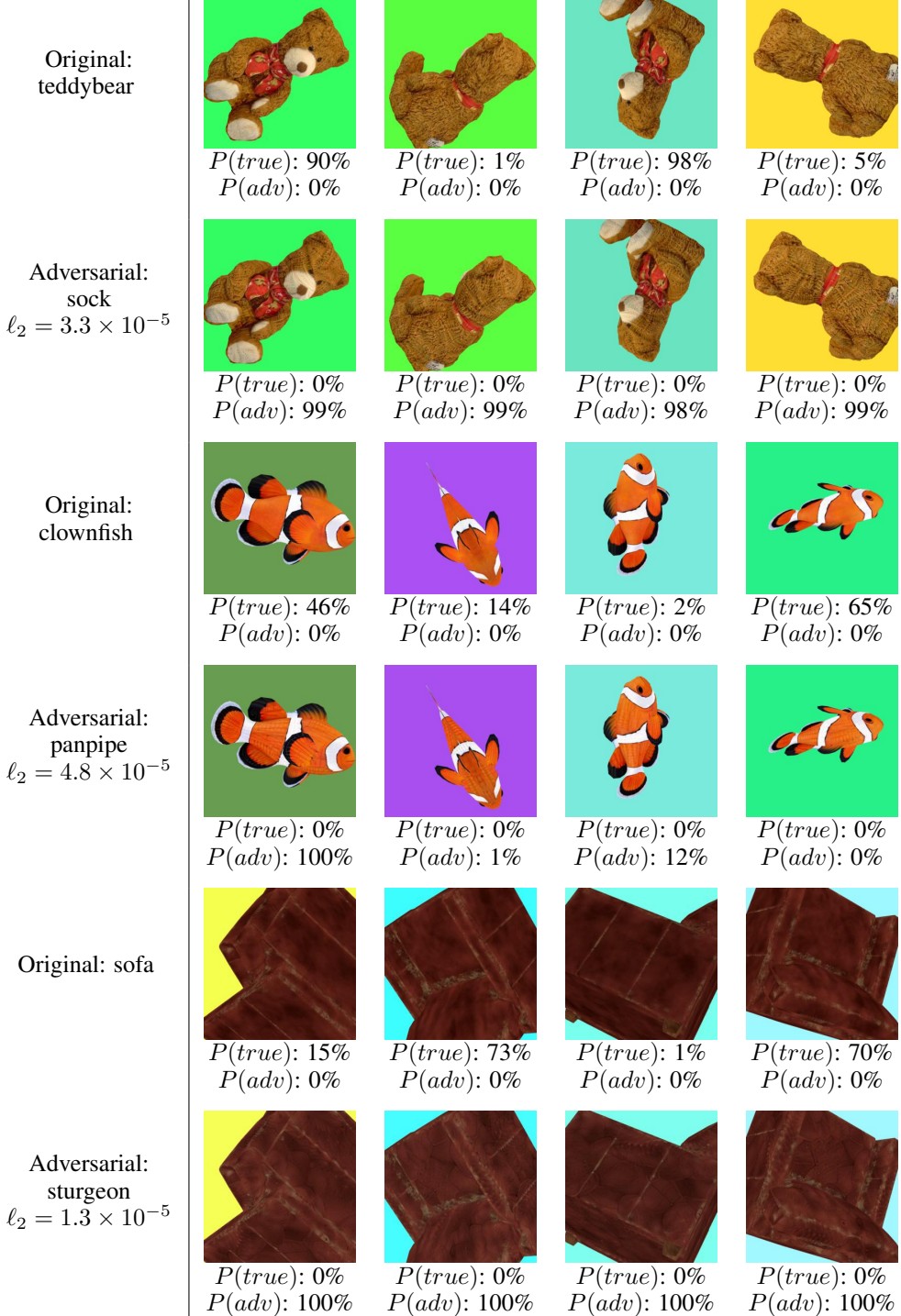

Figure 12: A random sample of 3D adversarial examples.

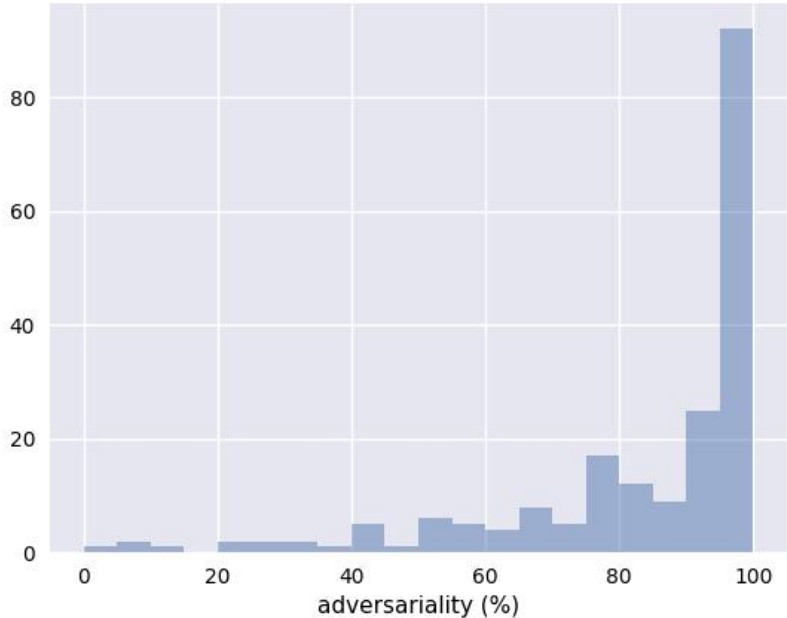

Figure 13: A histogram of adversariality (percent of 100 samples classified as the adversarial class) across the 200 3D adversarial examples.

## D   PHYSICAL ADVERSARIAL EXAMPLES

Figure 14 gives all 100 photographs of our adversarial 3D-printed turtle, and Figure 15 gives all 100 photographs of our adversarial 3D-printed baseball.

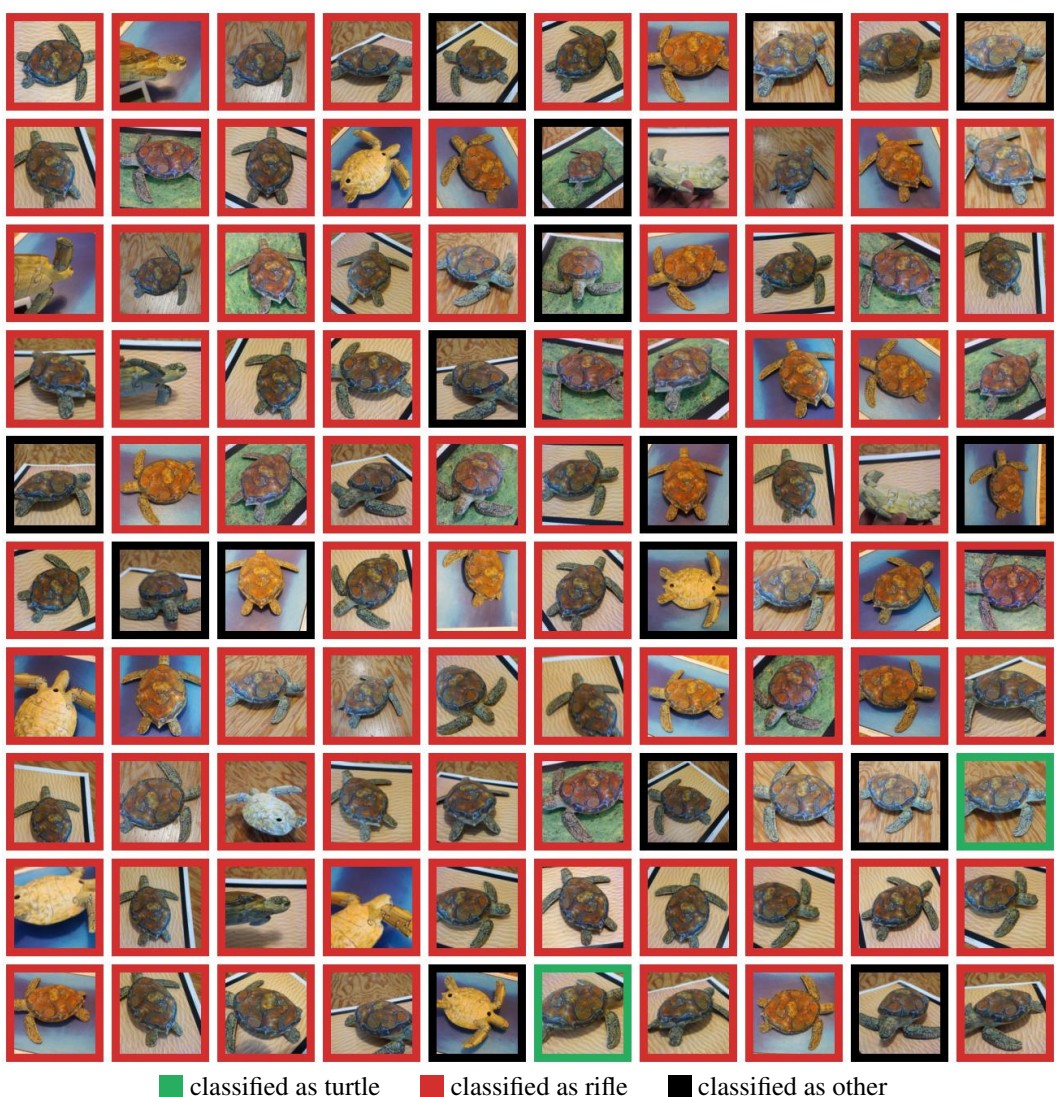

■ classified as turtle    ■ classified as rifle    ■ classified as other

Figure 14: All 100 photographs of our physical-world 3D adversarial turtle.

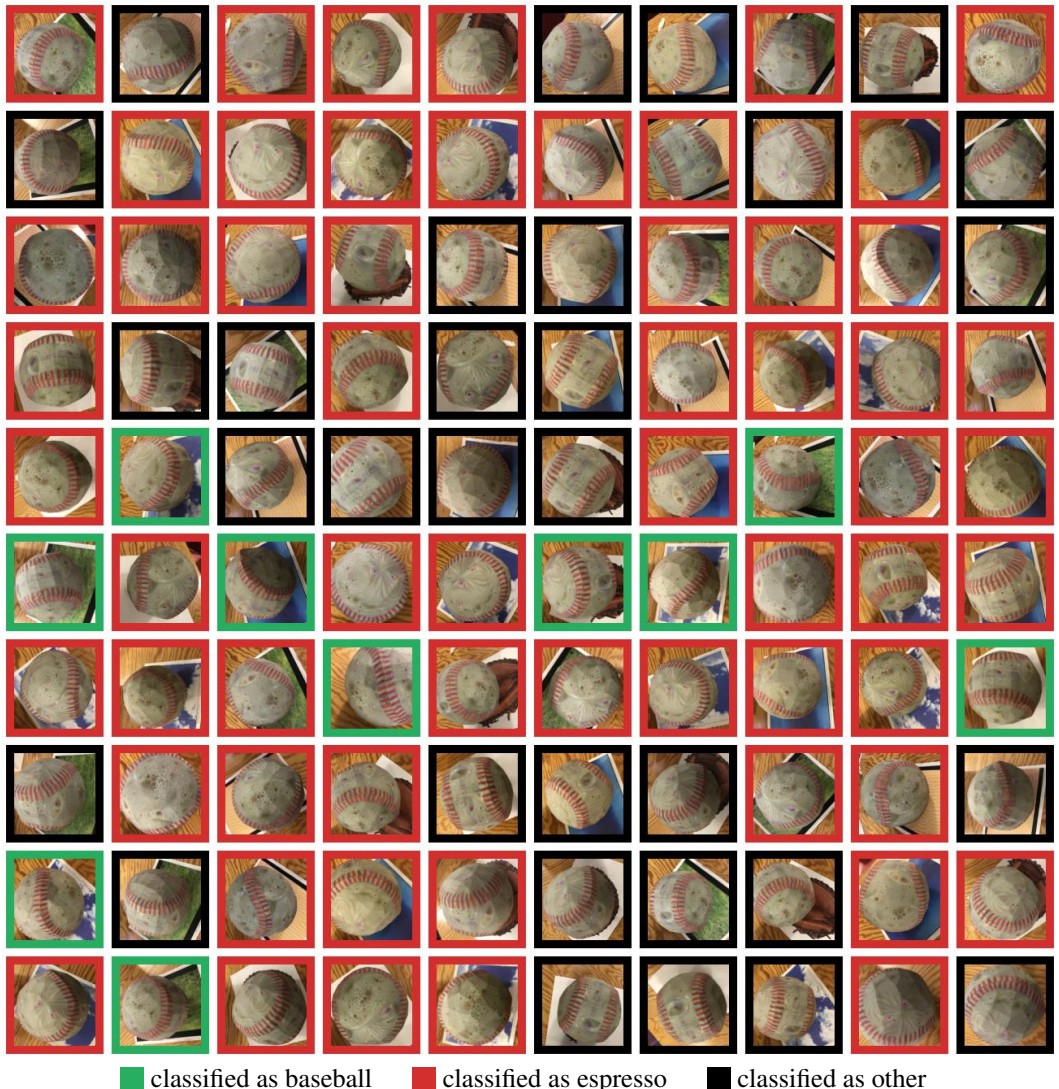

Figure 15: All 100 photographs of our physical-world 3D adversarial baseball.

