# OpenReview forum: "Synthesizing Robust Adversarial Examples"
_ICLR.cc/2018/Conference — Reject_

### Official Review · AnonReviewer2 · 2017-11-27
**Interesting idea but needs a major revision**

**Rating:** 5
**Confidence:** 4

**Review:**

Summary: This work proposes a way to create 3D objects to fool the classification of their pictures from different view points by a neural network.
Rather than optimizing the log-likelihood of a single example, the optimization if performed over a the expectation of a set of transformations of sample images. Using an inception v3 net, they create adversarial attacks on a subset of the imagenet validation set transformed by translations, lightening conditions, rotations, and scalings among others, and observe a drop of the classifier accuracy performance from 70% to less than 1%. They also create two 3D printed objects which most pictures taken from random viewpoints are fooling the network in its class prediction.


Main comments:
- The idea of building 3D adversarial objects is novel so the study is interesting. However, the paper is incomplete, with a very low number of references, only 2 conference papers if we assume the list is up to date.
See for instance Cisse et al. Houdini: fooling Deep Structured Prediction Models, NIPS 2017 for a recent list of related work in this research area.
- The presentation of the results is not very clear. See specific comments below.
- It would be nice to include insights to improve neural nets to become less sensitive to these attacks.


Minor comments:
Fig1 : a bug with color seems to have been fixed
Model section: be consistent with the notations. Bold everywhere or nowhere
Results: The tables are difficult to read and should be clarified:
What does the l2 metric stands for ?
How about min, max ?
Accuracy -> classification accuracy
Models -> 3D models
Describe each metric (Adversarial, Miss-classified, Correct)

---

> ### Author Response · Authors · 2018-01-05
> **Thank you**
>
> Thank you for your review. In the latest revision of our paper we greatly expand on the related work section, both discussing in more detail our current list, and introducing other related works from the field which we explain and differentiate from our own. We hope that this gives the reader a more complete view of the field, and further indicates the novelty of our work.
>
> The focus of this work was to demonstrate that it is possible to construct transformation-tolerant adversarial examples, even in the physical world; defenses against adversarial examples are beyond the scope of this paper. We hesitate to present intuitions for defenses without rigorous experimentation, because as researchers like Carlini have shown, developing defenses is challenging, and many proposed ideas for defenses are easily defeated [1].
>
> We have addressed all of the minor comments including:
> * Fixing the color bug
> * Removing the selected bolding from the model section
> * Elaborated and defined the l2 metric, and removed min/max in favor of mean/stdev
> * Models -> 3D models and accuracy -> Classification accuracy
> * Added a paragraph defining the terms “adversarial,” “misclassified,” and “correct” as we use them
>
> [1]: https://arxiv.org/abs/1705.07263

---

### Official Review · AnonReviewer3 · 2017-11-28
**Review - Accept pending clarifications**

**Rating:** 6
**Confidence:** 4

**Review:**

The authors present a method to enable robust generation of adversarial visual
inputs for image classification.

They develop on the theme that 'real-world' transformations typically provide a
countermeasure against adversarial attacks in the visual domain, to show that
contextualising the adversarial exemplar generation by those very
transformations can still enable effective adversarial example generation.

They adapt an existing method for deriving adversarial examples to act under a
projection space (effectively a latent-variable model) which is defined through
a transformations distribution.

They demonstrate the effectiveness of their approach in the 2D and 3D
(simulated and real) domains.

The paper is clear to follow and the objective employed appears to be sound. I
like the idea of using 3D generation, and particularly, 3D printing, as a means
of generating adversarial examples -- there is definite novelty in that
particular exploration for adversarial examples.

I did however have some concerns:

1. What precisely is the distribution of transformations used for each
   experiment? Is it a PCFG? Are the different components quantised such that
   they are discrete rvs, or are there still continuous rvs? (For example, is
   lighting discretised to particular locations or taken to be (say) a 3D
   Gaussian?) And on a related note, how were the number of sampled
   transformations chosen?

   Knowing the distribution (and the extent of it's support) can help situate
   the effectiveness of the number of samples taken to derive the adversarial
   input.

2. While choosing the distance metric in transformed space, LAB is used, but
   for the experimental results, l_2 is measured in RGB space -- showing the
   RGB distance is perhaps not all that useful given it's not actually being
   used in the objective. I would perhaps suggest showing LAB, maybe in
   addition to RGB if required.

3. Quantitative analysis: I would suggest reporting confidence intervals;
   perhaps just the 1st standard deviation over the accuracies for the true and
   'adversarial' labels -- the min and max don't help too much in understanding
   what effect the monte-carlo approximation of the objective has on things.

   Moreover, the min and max are only reported for the 2D and rendered 3D
   experiments -- it's missing for the 3D printing experiment.

4. Experiment power: While the experimental setup seems well thought out and
   structured, the sample size (i.e, the number of entities considered) seems a
   bit too small to draw any real conclusions from. There are 5 exemplar
   objects for the 3D rendering experiment and only 2 for the 3D printing one.

   While I understand that 3D printing is perhaps not all that scalable to be
   able to rattle off many models, the 3D rendering experiment surely can be
   extended to include more models? Were the turtle and baseball models chosen
   randomly, or chosen for some particular reason? Similar questions for the 5
   models in the 3D rendering experiment.

5. 3D printing experiment transformations: While the 2D and 3D rendering
   experiments explicitly state that the sampled transformations were random,
   the 3D printing one says "over a variety of viewpoints". Were these
   viewpoints chosen randomly?

Most of these concerns are potentially quirks in the exposition rather than any
issues with the experiments conducted themselves. For now, I think the
submission is good for a weak accept –- if the authors address my concerns, and/or
correct my potential misunderstanding of the issues, I'd be happy to upgrade my
review to an accept.

---

> ### Author Response · Authors · 2018-01-05
> **Thank you**
>
> Thank you for your review. We have made several clarifications in the exposition that we believe address your concerns and improve the paper. In particular:
>
> 1. The parameters of the distribution used in generating examples was given in the Appendix, but the method by which they are sampled was not made clear; we now explicitly state in the evaluation section that the parameters are sampled as independent uniformly distributed continuous random variables (except for Gaussian noise, which is sampled as a Gaussian continuous RV). There was no fixed number of transformations chosen during the synthesis of the adversarial example: the transformations are independently sampled at each gradient descent step. We have updated the text in the approach section to clarify this.
>
> 2. Yes, we agree: we minimized LAB, not RGB, and Euclidean distances make more sense in a perceptually uniform color space like LAB. We have switched to reporting LAB distances.
>
> 3. While we gave the distribution of adversariality across examples in a graph the appendix, we did not explicitly state the standard deviation/confidence intervals. This has been resolved in the latest version. We have also removed the min and max metrics from the evaluation section, and have added the standard deviation over the accuracies for the true and ‘adversarial’ labels as suggested. We report mean/standard deviation for 2D and rendered 3D experiments and not the 3D printing experiment because we report the statistics for each 3D objects separately.
>
> 4. In the case of the 3D printing experiment, we were limited by printer capability and shipping feasibility for this revision, but would be happy to include a few more in the camera ready version. We also included 5 more models in the 3D rendering experiment, making a total of 200 adversarial examples (10 models, 20 randomly chosen targets for each model). The turtle and baseball models were chosen because they could be easily adapted for the 3D printing process. The adversarial targets for the turtle and baseball (as well as all our other experiments) were randomly chosen across all the eligible ImageNet classes. Models for the 3D simulation experiment were chosen based on the first 10 realistic, textured 3D models we could find in OBJ format.
>
> 5. We have added a footnote to address this concern; although the viewpoints were not selected or cherry-picked in any capacity, we opt to not call them “random” because in contrast to the 2D and 3D virtual examples, the viewpoints were not (and realistically could not have been) uniformly sampled from some concrete distribution of viewpoints; instead the objects were repeatedly moved and rotated on a table with humans walking around them and taking pictures from “a variety of viewpoints.”

---

### Official Review · AnonReviewer1 · 2017-11-28
**Persuasive real-world results, would benefit from a comparison to universal examples**

**Rating:** 8
**Confidence:** 3

**Review:**

The paper proposes a method to synthesize adversarial examples that remain robust to different 2D and 3D perturbations. The paper shows this is effective by transferring the examples to 3D objects that are color 3D-printed and show some nice results.

The experimental results and video showing that the perturbation is effective for different camera angles, lighting conditions and background is quite impressive. This work convincingly shows that adversarial examples are a real-world problem for production deep-learning systems rather than something that is only academically interesting.

However, the authors claim that standard techniques require complete control and careful setups (e.g. in the camera case) is quite misleading, especially with regards to the work by Kurakin et. al. This paper also seems to have some problems of its own (for example the turtle is at relatively the same distance from the camera in all the examples, I expect the perturbation wouldn't work well if it was far enough away that the camera could not resolve the HD texture of the turtle).

One interesting point this work raises is whether the algorithm is essentially learning universal perturbations (Moosavi-Dezfooli et. al). If that's the case then complicated transformation sampling and 3D mapping setup would be unnecessary. This may already be the case since the training set already consists of multiple lighting, rotation and camera type transformations so I would expect universal perturbations to already produce similar results in the real-world.

Minor comments:
Section 1.1: "a affine" -> "an affine"
Typo in section 3.4: "of a of a"
It's interesting in figure 9 that the crossword puzzle appears in the image of the lighthouse.

Moosavi-Dezfooli, S. M., Fawzi, A., Fawzi, O., & Frossard, P. Universal adversarial perturbations. CVPR 2017.

---

> ### Author Response · Authors · 2018-01-05
> **Thank you**
>
> Thank you for the review and detailed comments. We are glad you enjoyed the paper. We have made revisions to the related work section, including a clearer description of Kurakin et al. and a more thorough discussion of other works (including the suggested “Universal Perturbations” paper by Moosavi-Dezfooli et al). We have additionally fixed all the minor issues you pointed out.

---

### Public Comment · ~Ian_Goodfellow1 · 2017-10-31
**Misrepresentation of Kurakin et al 2016**

This comment isn't a complete review and I won't make an accept / reject recommendation.

This comment is just a request to improve the description of the difference between this work and the work of Kurakin et al 2016.

1)
This submission says that the method from Kurakin et al "only works in carefully controlled environments." This is a direct contradiction of Kurakin et al, which states that the method works "without careful control of lighting, camera angle, distance to the page, etc."

The discrepancy probably results from different definitions of what "careful control" means. The language used in the paper should be more precise and specific in order to avoid seeming contradictory. We used considerably less control of the photograph conditions than standard protocols for commercial studio / event photography (which use special lighting equipment and camera tripods) or lab photography for scientific experiments.

It's also worth mentioning that we did a successful live demonstration of the method on stage at GeekPwn 2016, where the lighting, viewpoint, etc. were considerably less controlled than the experiments in the original paper (bright stage lights in a dark room, paper held in a presenters' hands instead of lying on a flat surface, etc.)

 I would suggest rewording to something like "Kurakin et al 2016 evaluated their method for approximately axis-aligned views in office lighting conditions and in stage lighting conditions at GeekPwn 2016. They used a hand-held camera to photograph the images from positions that are natural from a human user. This was controlled more than the work in the sense that the viewpoint and lighting were usually approximately the same but not high controlled, in the sense that the camera was handheld, the distance and angle were not measured, and no attempt was made to control the lighting to be more standard than normal office conditions."

2)
Without new experiments, it cannot be said that the method from Kurakin et al "only works" in those settings. It is accurate to say that Kurakin et al "only evaluated" their method in those conditions. Unless you've repeated our experiments with more diversity in viewpoint, you can't claim positively that the method doesn't work.

3)
This paper says "When generated with standard methods, these examples do not consistently fool a classifier in the physical world due to viewpoint shifts, camera noise, and other natural transformations." This is not quite true; Figure 6 of Kurakin et al shows that some of these transformations easily destroy adversarial examples while others have smaller effects.

---

> ### Author Response · Authors · 2017-11-02
> **Clarification**
>
> Thank you for the feedback. We’ve made the following changes for the revised version:
>
> 1. To resolve the misunderstanding caused by the term “controlled,” we’ve removed the word and, instead, directly stated the experimental conditions of Kurakin et al., as described in their Section 3.2 Experimental Setup. In their setup, a photo is taken of the image, then warped so that “each example has known coordinates” using printed QR codes, and cropped “so that they would become squares of the same size as source images.” We also describe the attached video directly as “approximately axis-aligned.” We write our analysis of Kurakin et al. based on the distribution of transformations described in the latest version of the peer-reviewed ICLR 2017 Workshop paper. Please advise if a better description of the setup is available. Note that our method covers any differentiable transformation, and our 2D experiments cover noise and lightening/darkening, as well as rotation, skew, translation, and zoom, the three of which are not covered in Kurakin et al. (our 3D experiments cover even more). We are sorry for the misunderstanding.
>
> 2. We will remove the world “only” and mention exactly the setup described in Kurakin et al as described above. We did experiments and they indicated that the Kurakin et al. method fails under a combination of rescaling, rotation, and translation (e.g. as used in the distribution used in our 2D case, see Table 4 in the Appendix); However, because robustness to such transformations was never claimed in Kurakin et al., we decided to not include these findings in our paper.
>
> 3. Our work states that in general, adversarial examples fail to transfer because of the combination of “viewpoint shifts, camera noise, and other natural transformations.” The degrees to which each of these transformations contribute was not studied, nor were any related claims made. We will make this more explicit in our revised version.
>
> We hope that these edits, in addition to the other differentiations already stated in the paper (untargeted vs targeted adversarial attacks, and 2D vs 3D examples) now appropriately represent the difference between Kurakin et al. and this work. We welcome additional feedback, and we thank you again for helping us improve our writeup.

---

> > ### Public Comment · ~Ian_Goodfellow1 · 2017-11-02
> > **Thanks**
> >
> > Thanks, that sounds like the revision fixes the issue.
> >
> > Regarding point 3, it sounds like you're going to fix it, but to explain why I made this comment: in my original comment, item 3, the sentence I quote doesn't contain a word like "combination" to disambiguate whether the reader is meant to parse the list as an OR or as an AND. From your reply it sounds like you're going to disambiguate this as AND.
> >
> > I hope it's clear that I wasn't saying your paper lacks novelty. I was just saying the Kurakin paper wasn't as limited as described. Overall I like your paper.

---

> > > ### Author Response · Authors · 2017-11-02
> > > **Disambiguated**
> > >
> > > Yes, we've disambiguated it to mean the combination of the natural transformations as you suggest. Thanks again for your feedback, and please let us know if you see anything else we can improve.

---

> > > ### Author Response · Authors · 2018-01-05
> > > **Paper updated**
> > >
> > > Our updated paper includes a revised abstract and related work that takes into account your feedback. We hope this clarifies our explanation of the work of Kurakin et al. Please let us know if you have any other feedback on the related work or the ideas presented in the rest of the paper, and we will take it into account before the next deadline.

---

### Public Comment · (anonymous) · 2017-11-02
**Robustness of Perturbation**

The images including the videos appear to be taken within a short distance from the object, I wonder if the distance will affect the perturbation. If so, what's the distance range within which the perturbation is robust.
Is such perturbation able to attack detection algorithm, such as YOLO and Fastrcnn?

---

> ### Author Response · Authors · 2017-11-02
> **Re: Robustness of Perturbation**
>
> EOT produces examples that are robust under the chosen distribution; it does not promise anything about out-of-distribution samples. We demonstrate that our adversarial examples work over varying levels of zoom (in addition to other transformation) in both the 2D and 3D cases: see our Appendix for the exact parameters we chose.
>
> Our research focuses on classifiers. We did not try attacking YOLO or Fast-RCNN in this work. However, given that detectors use a pretty similar architecture (and basically re-use the classifier, like VGG-16), we expect that it wouldn't be very different to attack a detector.

---

> > ### Public Comment · (anonymous) · 2017-11-04
> > **Re: Robustness of Perturbation**
> >
> > I agreed with the authors that generating robust adversarial examples to detection algorithm is possible. There is already one paper demonstrates the successful attack on faster-rcnn,
> >
> > Xie C, Wang J, Zhang Z, et al. Adversarial Examples for Semantic Segmentation and Object Detection
> >
> > maybe simply combine them can achieve this goal.

---

### Public Comment · (anonymous) · 2017-11-03
**Future work**

Validating your ideas on a 3D printed model is interesting.
What is the future direction of your work?

---

> ### Author Response · Authors · 2017-11-03
> **Re: Future work**
>
> Thanks! We think it would be neat to work on extending this to black-box systems and systems deployed in the real world.

---

### Public Comment · ~prabhant_singh1 · 2017-12-15
**Reproducibility Report**

The current report has been produced as a part of ICLR reproducibility challenge

Author: Prabhant Singh, University of Tartu, prabhant.singh@ut.ee

**Abstract:**
The paper’s main goal was to provide an algorithm to generate adversarial examples that are robust across any chosen distribution of transformations. The authors demonstrated this algorithm in 2 and 3 dimensions in the paper. The authors were successfully able to demonstrate that adversarial examples are a practical concern for real-world systems. During the reproducibility of the paper, we have implemented authors’ algorithm on the 2D scenario and were able to verify authors’ claim. We have also checked for transferability with the image of 3D adversarial example generated in this paper in the real-world environment. This report also checks the robustness of adversarial examples on black box scenario which was not in the selected paper.


**Experimental methodology:**
After reproducing the Expectation Over Transformation (EOT) algorithm we have generated adversarial examples on the pre-trained inceptionV3 model trained on ImageNet dataset. The adversarial examples were robust under the predefined distribution. One interesting observation here is that whenever we rotated the image out of the distribution there was confidence reduction in case of prediction and the target class which was predefined while creating the adversarial example was within the top 10 probabilities. The probability of target class was decreased when we rotated it away from the distribution and vice versa. As the paper states, there are no guarantees of adversarial examples being robust outside the chosen distribution but the adversarial example was still able to reduce the confidence of the prediction.


**Transferability:**
The transferability was checked on four images.  First image was generated by EOT and other three were of adversarial Turtle mentioned in the paper [1]. The transferability was tested on six different architectures pre-trained on the ImageNet dataset (Resnet50, InceptionV3, InceptionResnetV2, Xception, VGG16, VGG19). Our adversarial examples were generated using Tensorflow pre-trained Inception model. The transferability was checked with pre-trained keras models[2].
The results of the experiments are listed below:

Generated adversarial image using EOT
Parameters:
Learning rate: 2e-1
Epsilon: 8.0/255.0
True class: Tabby cat
Target class: Guacamole

1. InceptionV3:
Prediction: Flatworm, Confidence: 100%
2. InceptionResnet:
Prediction: Comicbook, Confidence : 100%
3. Xception:
Prediction: Necklace, Confidence : 92.5%
4. Resnet50:
Prediction: Tabby cat, Confidence: 35%
5. VGG 19:
Prediction: Tabby cat, Confidence: 47.9%
6. VGG16
Prediction: Tabby cat, Confidence: 34.8%

Image of 3D adversarial turtle[1] mentioned in the paper
True class: Turtle

1. InceptionV3:
Prediction: Pencil sharpner, Confidence : 67.7%
2. InceptionResnet:
Prediction: Comic book, Confidence : 100%
3. Xception:
Prediction: Table lamp, Confidence : 84.8%
4. Resnet50:
Prediction: Bucket , Confidence: 20%
5. VGG 19:
Prediction: Mask, Confidence: 10.9%
6. VGG16
Prediction: Turtle, Confidence: 3.6%

Other images of Adversarial turtle generated similar results.

**Observations:**

Both images of adversarial turtle and cat were detected incorrectly by inception related architectures with a high confidence.
Both images were classified as “Comic book” with 100 percent confidence by InceptionResnetV2.
The adversarial examples were able to reduce the confidence by a high margin, about 50-60 percent in case of Tabbycat. Only VGG16 was able to classify the turtle correctly but by a very low confidence of 3.6%
Similar results were found when we rotated, cropped and zoomed out of the image.[3]
In case of adversarial turtle, the photo was taken out of the distribution(Not inside the chosen distribution as mentioned in the paper ie camera distance between 2.5cm -3.0cm) ,still the image was misclassified.

**Conclusion:**

The author successfully generated robust adversarial examples which are robust under the given distribution in case of targeted misclassification. The adversarial examples were also robust in case of untargeted misclassification under any distribution if classified against Inception related models.The adversarial examples reduced confidence by a wide margin in case of non-inception architectures. The image of 3D adversarial turtle can be considered robust under any distribution as it has been misclassified against all the architectures and only classified correctly by VGG16 but with a very insignificant percentage.

**Sources:**
[1] The Image of the adversarial turtle was taken at the recent NIPS conference by a number of viewpoints out of the given distribution.

[2] Pre-trained keras models: https://keras.io/applications/

[3] The source code and experiments info can be found in this Github repo: https://github.com/prabhant/synthesizing-robust-adversarial-examples

---

> ### Author Response · Authors · 2018-01-05
> **Re: Reproducibility Report**
>
> Thank you for taking the time to reproduce the results in our paper! We’re glad that you were able to replicate our results.
>
> We assume that you did not reproduce our 3D results due to a lack of an openly available differentiable renderer (it is somewhat of a pain to implement). We’ll do our best to have ours open-sourced by the time of the conference.
>
> It’s interesting to see that there’s some degree of transferability between these EOT adversarial examples as well; we hadn’t explored this much in our work. If you do explore this further, or try new things like optimizing over an ensemble, please let us know how it goes!

---

> > ### Public Comment · ~Kranthi_Kiran_GV1 · 2018-01-24
> > **Suggest directions to take it further**
> >
> > Can you comment on what aspects can be explored further?

---

### Author Response · Authors · 2018-01-05
**Update**

We thank the anonymous reviewers for helping us improve the paper. In response to their feedback, we have made the following revisions to our paper (in addition to the fixing of a few spelling mistakes/typos):

* Related work: We have updated our description of Kurakin et al. and added a comparison with Universal Adversarial Perturbations (Moosavi-Dezfooli et al.) and adversarial eyeglasses (Sharif et al.)

* Evaluation: We have included an additional 5 models in our Robust 3D Adversarial Examples evaluation. We have additionally further explained our evaluation metrics, improved and defined previously confusing terminology, and added standard deviations to further elucidate the distributions of adversariality and classification accuracy.

---

### Decision · Program_Chairs · 2018-01-29
**ICLR 2018 Conference Acceptance Decision**

**Decision:**

Reject

**Comment:**

This paper studies the problem of synthesizing adversarial examples that will succeed at fooling a classification system under unknown viewpoint, lighting, etc conditions. For that purpose, the authors propose a data-augmentation technique (called "EOT") that makes adversarial examples robust against a predetermined family of transformations.

Reviewers were mixed in their assessment of this work, on the one hand highlighting the potential practical applications, but on the other hand warning about weak comparisons with existing literature, as well as lack of discussion about how to improve the robustness of the deep neural net against that form of attacks.
The AC thus believes this paper will greatly benefit from a further round of iteration/review, and therefore recommends rejection at this time.